# Quantifying Visual Pollution from Urban Air Mobility

**Kilian Thomas** [1] and **Tobias A. Granberg** [2,*]

1   Airports and Air Transport Specialization, Escuela Técnica Superior de Ingeniería Aernáutica y del Espacio (ETSIAE), Universidad Politécnica de Madrid, 28040 Madrid, Spain; kilian.thomas@alumnos.upm.es
2   Division of Communications and Transport Systems, Linköping University, 601 74 Norrköping, Sweden
*   Correspondence: tobias.andersson.granberg@liu.se

**Abstract:** Unmanned aerial vehicles (UAVs) can bring many benefits, particularly in emergency response and disaster management. However, they also induce negative effects, such as noise and visual pollution, risk, and integrity concerns. In this work, we study visual pollution, developing a quantitative measure that can calculate the visual pollution from one or multiple UAVs. First, the Analytic Hierarchy Process was utilized in an expert workshop to find and rank factors relevant to visual pollution. Then an image-based questionnaire targeted at the general public was used to find relations between the factors. The results show that the two main factors causing visual pollution are the number of UAVs and the distance between a UAV and the observer. They also show that while a UAV used for emergency medical services is as polluting as any other UAV, it is easier to tolerate this pollution. Based on the questionnaire results, two visual pollution functions were developed that can be used when carrying out path planning for one or multiple UAVs. When combining this function with other existing measures for noise pollution, and ground and air risk, it is possible to find paths that will give as little negative impact as possible from urban air mobility.

**Keywords:** visual pollution; UAV; UAM; eVTOL; drones; AHP; path planning; route planning

## 1. Introduction

Unmanned aerial vehicles (UAVs) and electric vertical take-off and landing (eVTOL) aircraft (both often called drones) are expected to be an integral part of the transportation system in the future, contributing to efficient and environmentally friendly transportation of both freight and people [1]. To achieve this, they must be automated (possible to fly beyond visual line of sight) and there must exist an unmanned traffic management (UTM) system that ensures safe operations [2]. While various organizations and companies are working towards establishing this UTM system allowing large-scale operations, tests are currently being carried out for different applications, including using UAVs as an emergency response resource [3]. In an emergency response and disaster management context, UAVs are already being tested or used for the delivery of automated external defibrillators [4,5], to help find missing persons [6], find drowning victims [7], obtain aerial images of disaster sites [8], and detect dangerous chemicals [9], etc.

Among the challenges of implementing UAVs and eVTOLs as part of Urban Air Mobility (UAM) are the negative effects that they may be perceived to produce. Two such effects are noise and visual pollution. It is well known that noise exposure may cause health issues [10], and studies have shown that noise from UAVs is more annoying than from other transportation sources [11]. This is expected to be true also when considering visual pollution, where people would get more annoyed to see a passing UAV than a passing car [12].

While noise pollution has been extensively studied (e.g., [13–17]), this is not the case for visual pollution. "Visual pollutants" [18] may refer to advertisements, signage, and litter, but also any other element—both indoor and outdoor—that an observer finds unpleasant or offensive to look at. They also degrade the visual quality of a place [19]. Thus, visual

pollution is the negative impact that the view of some artificial structure or object (visual pollutant) and its movement might have on a person. UAVs and eVTOLs are examples of possible visual pollutants.

One of the first definitions for visual pollution is "The degradation of the visual quality of historic city centres caused by commercial signs displayed on building facades and in public spaces" [20]. Two years later, [21] claimed that "Visual pollution is a designation broadly employed to cover limits on the ability to view distant objects, to describe the subjective issues produced by the introduction of structures in beautiful scenes, and as a way to refer to all other visual defacements". In [22], the authors stated that visual pollution was a term to express how the introduction of negative changes might disturb people, to which [23] added that visual pollution may increase if the pollutant is moving because it attracts people's attention and can reduce peacefulness.

In [24], a list of all aspects that may be considered visual pollutants is provided: "Reckless placing of stickers, waste thrown in random places or in front of residential houses, misplaced containers, buildings that are not in harmony with the surrounding infrastructure, bad urban planning, streets lands that are not homogeneous, protruding buildings, irregularity, parking spots with a clear lack of order, communication towers, antennas, wires and advertising ( . . . )". They highlight that visual pollution is not only about the beauty of objects but is highly interconnected to the spatial arrangement of those objects and whether or not they are well-organized. This interpretation is also made by [25], who establishes that visual pollution is spatial chaos.

The interest in studying visual pollution increased with the introduction of wind farms, which annoy parts of the population simply by their presence. Different studies have attempted to quantify their impact [23,26,27]. The study by Ref. [24] considered that some objects, such as wires, rubbish, or landfills, will always be regarded as visual pollutants, and forms of classifying different types of visual pollutants are suggested by [19,28].

Visual pollution may cause health issues, such as [24]:

- Distraction and lack of focus
- Stress and anxiety
- Difficulty in processing visual input due to the extensive amount of simultaneous data
- Dangerous distractions, especially in a driving context
- Reduced work efficiency
- A low frame of mind
- Mood disorders and aggression

From an economic perspective, visual pollution may induce indirect costs or property value loss. In the Czech Republic, 85% of the proposed wind turbine projects have been aborted because of this [26]. In Greece, it was found that the sales price of the per-unit floor in the areas that were nearby a wind farm location decreased because of the visual impact [27].

Visual pollution is not a problem that disappears with time. In [29], for example, 60 out of 72 participants (83%) answered that their city was visually polluted, even if they were obviously used to their living environment. Furthermore, this problem also increases the perception of other problems. For instance, noise pollution is amplified if the source generating the sound is visible [30].

However, visual pollution does not seem to be among the top concerns when implementing UAM. EASA survey results [31] show that visual pollution is the second to last factor (out of ten) concerning citizens when considering UAV deliveries, ranking less important than safety and noise pollution. However, the report remarks that it is not to be considered negligible. Another survey study showed that 22% of the participants claimed to be concerned that UAVs might make the sky less pleasant to look at [32]. However, visual pollution was the primary concern for only around 4% of the respondents.

Visual pollution has been previously studied using a range of different methods. The study by [28] provides an extensive summary of techniques that have been used, including color photographs, surveys, SWOT analysis, GIS tools, statistical analysis, etc.

Survey methods are commonly used in visual pollution studies, and numerous examples can be found [20,26,31]. Deep learning was used to classify and measure the visual pollution in Bangladesh [19], providing information about if there was visual pollution in a picture. Another similar approach was adopted in [18] using Google Maps tools and You Only Look Once (YOLO) methodologies. They claim that if the location of each photograph is captured, a map can be constructed with the type and significance of visual pollution present in each geographical zone.

The Hedonic Pricing Model is "a statistical method to estimate monetary value on a set of characteristics of a good, typically housing" [33]. This method has been widely employed to estimate the impact that a certain characteristic (the study variable) can have on the price of a house. This method was adopted in a visual pollution study [27] where they sought to analyze the effect the view of a wind turbine might have on housing prices. Using regression [34], they conclude that it may cause the price to drop by more than 14%.

In [34], ArcGIS was employed to generate a 2.5D model where the surface of a city in Poland was rendered. Then, an intervisibility analysis was carried out to determine how many outside advertising items could be seen from a set of selected observation points. This process was followed by a questionnaire executed in the previously mentioned points, in which respondents were asked if the visual pollution was annoying. Statistical analysis was used to provide the maximum number of billboards that a standard citizen would categorize as not "too annoying".

The tangential approach estimates visual pollution during a period, in contrast to the previously described instant measures. The study by [25] introduced a time dimension to visual pollution studies by introducing several pictures that emulated the path a person would follow in a city. Then the author could estimate how many views the passer-by was missing because of the presence of outside advertising.

The study by [28] used the Analytic Hierarchy Process (AHP) methodology first to create a list of all items that could be considered pollutants and then, with a group of twenty experts, categorize them. Following the AHP method, a comparison and ranking of different visual pollutants were obtained. For instance, open dumps of solid waste were classified as the most annoying pollutant, followed by billboards, etc.

Given the abstract and subjective nature of visual pollution, it is a difficult area to study, which may be why the previous work done on how UAVs and eVTOLs contribute to visual pollution is scarce. In particular, there has been no previous study on how visual pollution can be quantified, which is helpful if it is to be considered in, e.g., path planning or infrastructure planning.

Thus, the aim of this paper is to describe how visual pollution from UAM can be quantified, and how the resulting measure can be used. One additional focus is on the application area of Emergency Medical Services (EMS), i.e., when the UAV is used for transporting medical equipment, medicine, or personnel to an incident site. In the rest of the paper, we will use the term UAV for both UAVs and eVTOLs, i.e., even if passengers are transported, it will be referred to as a UAV.

This work is part of AiRMOUR (https://airmour.eu/, accessed on 5 June 2023), an EU-funded project supporting sustainable air mobility via emergency medical services. Compared to previous work, we:

- Identify and rank factors that contribute to the visual pollution produced by UAVs.
- Test the significance and establish the relationship between a selection of the identified factors.
- Construct a numerical visual pollution model that can be used to calculate the visual pollution produced by one or several UAVs.

In the next section, we describe the methods used to collect and analyze data. This is followed by a presentation of the results in Section 3 and the discussion in Section 4. Finally, conclusions and future research directions are presented in Section 5.

## 2. Materials and Methods

The methodology used in this work was based on the previous research described in Section 1 and is illustrated in Figure 1.

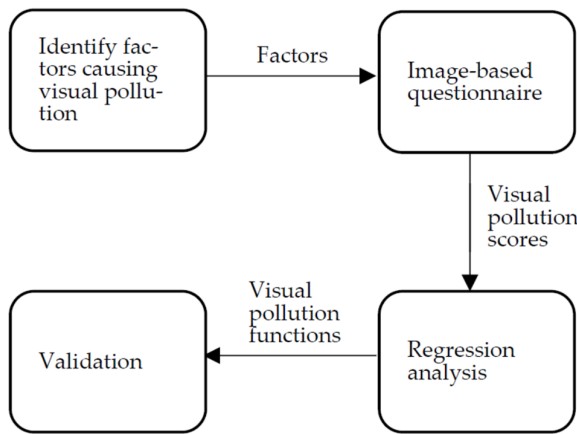

**Figure 1.** Methodology for the study.

The methodology consists of four main steps, briefly outlined below, and explained in more detail in the following subsections:

1.  Identifying the main factors that may cause visual pollution. An expert workshop was held to determine which factors to consider when analyzing visual pollution. During the workshop, the Analytic Hierarchy Process (AHP) methodology [35] was used to select the most important factors to include when constructing a mathematical model for calculating visual pollution. This gave a set of factors that might contribute to visual pollution from UAVs. A subset of these factors was then tested using an image-based questionnaire.
2.  An image-based questionnaire. A survey was crafted by adding UAVs in different positions and configurations to two backgrounds, one city, and one rural background. This is to see how the different identified factors would affect the perception of citizens. Then, respondents selected mainly by snowball sampling, were asked to rank the visual pollution in each image on a scale from zero to ten.
3.  Regression analysis. Regression was used to establish a correlation between the different factors and the grade that was obtained through the questionnaire. The objective was to obtain a function that could calculate the visual pollution based on different input values. Based on the results, two different functions were developed.
4.  Validating the results. During the test, four images were ranked by the participants but not considered in the regression, i.e., the images were divided into a training set and a validation set. Using these, it was possible to confirm that the functions can also be used for other scenarios.

### 2.1. Identifying the Main Factors That May Cause Visual Pollution

An expert workshop with seven participants was used for identifying and ranking factors on 24 March 2022. Six out of seven participants were part of the AiRMOUR project, all were very familiar with UAVs and UAM. The seventh participant had expertise in Air Traffic Management. The meeting mainly aimed to give input into which factors to include in the questionnaire, since including all possible factors would give too many questions.

At the start of the workshop, the participants were introduced to the topic, and a general discussion and brainstorming session on visual pollution from UAVs was held.

The experts were asked about the factors that, in their opinion, should be considered when building a model that can analyze visual pollution. The factors were grouped, reducing the number of independent parameters.

With a list of ten independent factors, the AHP began. AHP is a Multi-Criteria Decision-Analysis (MCDA) method that can be used to analyze and evaluate complex decisions [36]. Having a set of criteria, AHP lets decision-makers do pairwise comparisons, often using a scale from 1 to 9 [35]. For each criterion, numerical values are calculated based on the comparisons. AHP is only one of multiple MCDA methods (see, e.g., [37,38]). The reason for using it here was mainly because it had been successfully used before when studying visual pollution by [28], whose method was followed closely. Thus, the experts in the workshop were requested to fill in a questionnaire where they had to compare all the factors in pairs using a Saaty scale from 1 to 9.

When all participants finished their comparisons, they were requested to return the information. To get an aggregate response, there are at least two methods [39]. The first was to make a common matrix of preferences combining all participants' matrices and then obtain a single solution vector. The second method was to get the individual ranking for every expert and then combine the results. The last option was selected as it provided more individualized information about each expert. All the participants were assigned the same weight (same importance), and therefore, the final result was the geometric mean of each characteristic [35,39].

Finally, a study was made to find out if the experts had similar opinions. This was achieved with the Shannon entropy alpha and beta approach [40], giving a consensus indicator.

### 2.2. An Image-Based Questionnaire

Based on [19,20,26,34] among others, the main approach selected for establishing the relationship between the identified factors and visual pollution was an image-based questionnaire. It contained 3 general questions and 28 images. The format of all questions was closed [41] in favor of (1) getting numeric data, (2) reducing the time for completing the questionnaire, and (3) conforming to a uniform set of data for the analysis.

A 0 to 10 scale was selected, from "No pollution" to "Strong pollution". This choice of scale implied that only negative effects were reflected, not offering the chance of selecting a positive attitude further than "No pollution".

On the first page, interviewees were introduced to the subject and asked general information with three questions related to age, opinion about UAVs, and knowledge of UAVs. On the second page, the images with questions were presented (two example images are shown in Figures 2 and 3. The full survey is available as Supplementary File S1 in the Supplementary Material.

Each picture was followed by two questions:

1.    What is the level of visual pollution in this image?
2.    Is this level of visual pollution tolerable?

The questions were asked in a specific order. The first image showed a picture of a rural environment without any UAVs. The second image introduced one UAV in the same environment. This was to make sure that the respondents understood the main idea of the questionnaire. This structure was repeated in an urban setting in the third and fourth images. Then, pictures containing different configurations of UAVs at different distances were introduced randomly to avoid the growing effect [26]. Finally, images containing EMS UAVs and extra information were placed at the end of the questionnaire. All images were different, ensuring that interviewees would not feel unsure if they had already answered them [42]. Furthermore, the respondents were allowed to go backward and modify their ranking.

The questionnaire was designed to be as short as possible to avoid respondents quitting before finishing. Based on recommendations in [43–45], the whole survey was designed to take about 7 min; this resulted in 28 images.

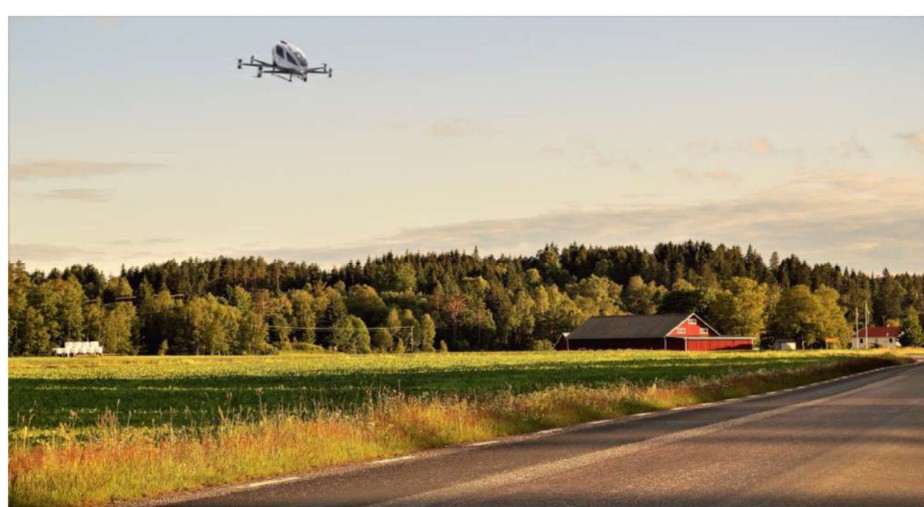

**Figure 2.** Questionnaire image with one UAV in a rural setting. (Background image from Pixabay, House in the field; uploaded by Grizzlybear-se, published 5 July 2017. Drone image from Ehang Scandinavia).

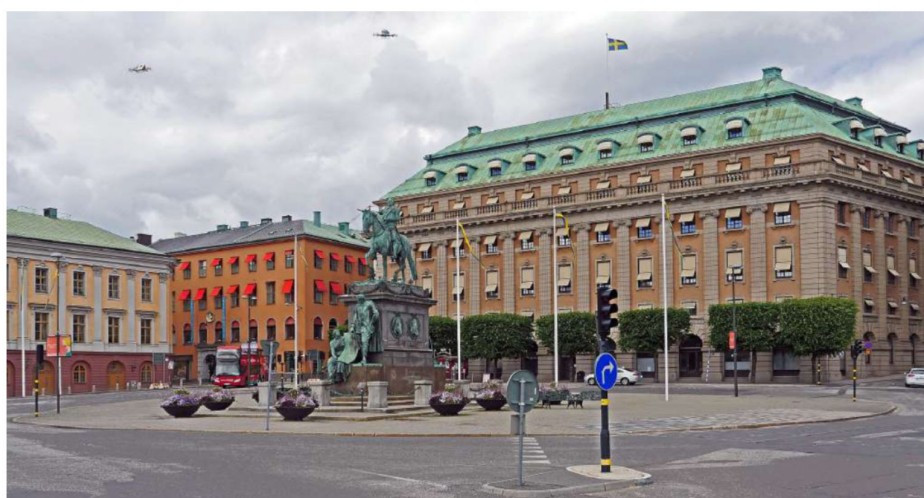

**Figure 3.** Questionnaire image with two UAVs in an urban setting. (Background image from Pixabay, Gustav Adolf Torg; uploaded by hpgruesen, published 5 July 2017. Drone image from Ehang Scandinavia).

The questionnaire was constructed with the Survey & Report platform from Artologik (https://www.artologik.com/en/survey-report, accessed on 5 June 2023). No specific selection of the respondents was made. We wanted general citizen participation, so the foremost approach for distributing the survey was snowball sampling. Colleagues, friends, and relatives were asked to fill out and spread the survey. This was complemented by website postings (e.g., on http://airmour.eu), email lists, and social media postings on, e.g., LinkedIn, Instagram, Reddit, and Facebook. The survey was available in English, Spanish, French, Swedish, German, Dutch, Norwegian, and Finnish and was open for 20 days.

*2.3. Regression Analysis*

In the data analysis, Excel was used to group and organize the information and carry out non-linear regression. Statgraphics was used for ANOVA tests and linear regression. Initial validation of the data was carried out with Excel. Histograms, maximum and minimum values, and general checks were performed, which indicated that some results were not logical and that some respondents may not have filled in the survey seriously. To identify these, several red flags (RF) were identified:

- RF1: The respondent ranked pictures without UAVs with a mark greater than or equal to 8.
- RF2: The respondent stated that images without UAVs were not tolerable.
- RF3: The respondent had 3 or more clear inconsistencies, e.g., that an image with a rank greater or equal to 8 was tolerable.
- RF4: The participant did not correctly identify the less polluted images. A number of pictures were, on average, low-ranked, and if a participant gave those a higher level than their own average, it was considered an inconsistency. If four or more images were inconsistently ranked, RF4 became active.
- RF5: The participant did not correctly identify the more polluted images. Same as RF4, but for the more polluted images.
- RF6, RF7, RF8, and RF9: The respondent provided responses not in line with the majority, e.g., the level of visual pollution increases when the UAV is further away.
- RF10: The respondent gave the same rank to all the images.
- RF11: The respondent gave more than six "10" ranks.

The responses of respondents that had four or more RFs were investigated further. This corresponded to 24 surveys. After a closer look, 3 of them turned out to be acceptable, and as a consequence, only 21 responses had to be dismissed. A total of 227 questionnaire responses from 248 remained for the analysis. The full set of responses to the questionnaire can be found in Supplementary File S2 in the Supplementary Material.

Once the survey data was cleaned, ANOVA tests were performed to check if there were significant differences among groups based on the participants' age, opinion about UAVs, and level of knowledge.

For the regression analysis, the variables included were classified according to:

- Picture: Number from 1 to 28 that represents the order in which the picture was shown in the test.
- Environment: 0—Rural, 1—Urban.
- Distance to the closest UAV.
- The number of UAVs.
- Purpose: 0—No EMS, 1—EMS (if it is an EMS UAV or not)
- Awareness: 0—There is no extra information in the picture. 1—There is extra information provided in the picture.
- ID: Respondent number from 1 to 248.
- Age: 1—Less than 18, 2—From 18 to 30, 3—From 30 to 50, 4—From 50 to 65, 5—More than 65.
- Opinion: 0—No opinion about UAVs, 1—Positive opinion, 2—Negative opinion.
- Knowledge: 1—None, 2—Basic (The participant stated that (s)he could write a paragraph about UAVs), 3—Medium (The participant could write a page about UAVs), 4—Expert (The participant could write five pages about UAVs).
- Rank: Number from 0 to 10, where zero is no pollution, and ten represents strong pollution.
- Tolerable: 0—Not tolerable, 1—Tolerable.

The data was first validated using histograms, qq graphics, Kolmogorov-Smirnov test, asymmetry and kurtosis tests, Levene's test, and the Durbin Watson test [46]. Then, several different types of regression models were tested in Statgraphics and in Excel, to provide a function that would fit the data as good as possible.

## 3. Results

### 3.1. Expert Workshop and Image-Based Questionnaire

As a result of the AHP-supported workshop, the list of factors in Table 1 was established as important when determining the visual pollution from UAVs.

**Table 1.** Factors selected during the expert workshop.

| Factor | Criteria | Description |
| --- | --- | --- |
| Appearance | Dimensions<br>Lights (static or flashing)<br>Color or icons | Observable characteristics of the UAV |
| Awareness | Knowledge about the UAV's route<br>Familiarity with UAVs<br>Trust in the application | Information (e.g., through an app) about the UAV, such as where it is going, where it comes from, its speed, etc. |
| Distance | Distance<br>Altitude | Distance to the observant and altitude of the UAV |
| Environment | Environment<br>What is it compared with? | In which environment the UAV is seen: in the city center, in a rural area, at the beach, etc. |
| Formation | Formation | If the UAVs are flying in a line formation, in groups, completely scattered, etc. |
| Movement | Movement | If the movement of the UAV or its speed has an influence on how it is perceived |
| Noise | Noise | If the noise generated by the UAV increases the visual pollution that it generates |
| Number of UAVs | Number of UAVs | How many UAVs are visible at the same time |
| Purpose | Purpose | If the UAV is carrying cargo, passengers, or is on an EMS mission |
| Temporal component | Pattern<br>Time of exposure<br>Frequency<br>Trajectory | Factors related to the time of exposure or the number of times that a person would see the UAV |

After establishing the list, the participants ranked the most important factors. Their results are summarized in Figure 4. The consensus range was 68%, and the inconsistencies were fairly high (up to 40% in one case), which means that the results have to be interpreted carefully.

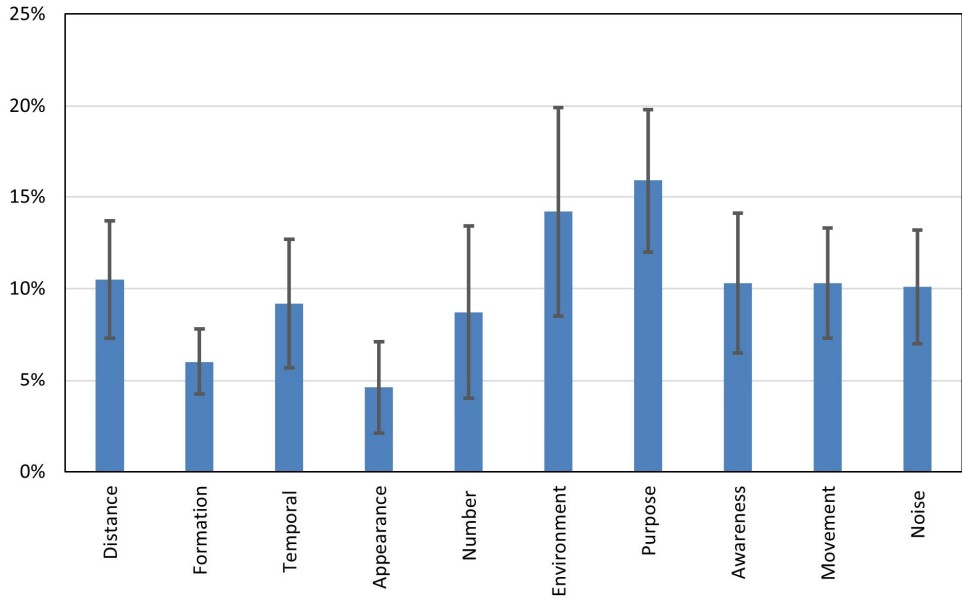

**Figure 4.** Importance of the factors, according to the AHP results.

3.1.1. Factors Included in the Questionnaire

The questionnaire was designed to capture the following factors:

- Purpose: Purpose was considered as a main factor affecting visual pollution and was thus included in the questionnaire, using two categories: EMS purpose, or not. To introduce the variable, EMS UAVs were colored green and shown in the introduction to make sure that every participant understood the meaning of this color code.
- Environment: It is reasonable to assume that the area in which a UAV is seen might have an influence on the visual pollution score. Thus, two different backgrounds were used, one urban and one rural.
- Distance: To study the effect of the distance between the observer and the UAV, the same UAV was placed at different distances: 80 m, 160 m, 320 m, and 640 m, following a similar approach as [26,34].
- Awareness: This factor tried to capture the possibility that a UAV might be less of a pollutant if the viewer knows where it is going, its speed, where it comes from, etc. To examine this, four existing images were complemented with information about the UAV's route, track, weight, speed, flight level, and a hypothetical owner company.
- Number of UAVs: Images with one, two, five, and ten UAVs were added to the questionnaire to investigate how the number of UAVs seen at the same time affects visual pollution.

3.1.2. Factors Not Included in the Questionnaire

The following factors were not added to the questionnaire:

- Movement: Although how the UAVs move might have a major influence on visual pollution, due to the static image-based questionnaire methodology, it could not be studied.
- Noise: Some studies (e.g., [30]) already concluded that viewing a noise's source might increase the noise annoyance. Nevertheless, and for the same reason as for movement, noise was not considered.
- Temporal factors: Visual pollution is not just an instant measure. It is probably more annoying to see a UAV for one minute than one hour. However, this factor was also deemed too difficult to capture using the selected methodology.
- Formation and appearance: According to the results from the AHP rankings, these two factors were perceived as minor when measuring visual pollution. Consequently, only one kind of UAV was used (an EHang 216), and when there were multiple UAVs, they had no specific formation.

The characteristics of each image can be seen in Table 2. The full survey can be seen in Supplementary File S1 in the Supplementary Material.

**Table 2.** Factors examined in each image in the questionnaire.

| Question | Environment | Distance to the Closest UAV | Number of UAVs | Purpose | Awareness |
|---|---|---|---|---|---|
| 1 | Rural | | 0 | | |
| 2 | Rural | 80 | 1 | No EMS | No |
| 3 | Urban | | 0 | | |
| 4 | Urban | 80 | 1 | No EMS | No |
| 5 | Urban | 160 | 1 | No EMS | No |
| 6 | Urban | 160 | 10 | No EMS | No |
| 7 | Rural | 80 | 2 | No EMS | No |

**Table 2.** *Cont.*

| Question | Environment | Distance to the Closest UAV | Number of UAVs | Purpose | Awareness |
|----------|-------------|-----------------------------|----------------|---------|-----------|
| 8 | Rural | 640 | 1 | No EMS | No |
| 9 | Urban | 320 | 1 | No EMS | No |
| 10 | Rural | 160 | 1 | No EMS | No |
| 11 | Urban | 80 | 2 | No EMS | No |
| 12 | Rural | 160 | 5 | No EMS | No |
| 13 | Urban | 320 | 2 | No EMS | No |
| 14 | Rural | 80 | 3 | No EMS | No |
| 15 | Rural | 320 | 1 | No EMS | No |
| 16 | Rural | 160 | 10 | No EMS | No |
| 17 | Urban | 640 | 1 | No EMS | No |
| 18 | Urban | 160 | 5 | No EMS | No |
| 19 | Rural | 160 | 2 | No EMS | No |
| 20 | Urban | 160 | 2 | No EMS | No |
| 21 | Rural | 80 | 1 | EMS | No |
| 22 | Rural | 160 | 1 | EMS | No |
| 23 | Urban | 80 | 1 | EMS | No |
| 24 | Urban | 160 | 1 | EMS | No |
| 25 | Urban | 80 | 1 | No EMS | Yes |
| 26 | Urban | 160 | 1 | No EMS | Yes |
| 27 | Rural | 80 | 1 | No EMS | Yes |
| 28 | Rural | 160 | 1 | No EMS | Yes |

*3.2. Results from the Questionnaire*

3.2.1. General Analysis of Respondents

From the 227 responses, only 3 belonged to participants who were less than 18 years old or older than 65. This meant that it was not possible to provide significant information about those age groups. A total of 53% of the respondents were between 18 and 30 years old, 25% were between 30 and 50 years old, and 18% were between 50 and 65 years old. Regarding opinion, almost 80% of the surveyed population had a positive opinion about UAVs, and only 14% had a negative opinion. The opinions were evenly distributed, meaning there was no specific group that tended to have better or worse opinions about UAVs. Concerning knowledge, most of the participants (also 80%) claimed to have a basic or medium knowledge of UAVs; however, 11% of the respondents classified themselves as experts and 9% claimed to have no knowledge at all about UAVs.

3.2.2. Comparisons among Different Groups of Respondents

Age did not appear to be a significant factor when analyzing visual pollution. After removing the answers of the youngest and the oldest groups due to the lack of participants, it was established that the means between all the other groups were almost identical and that there was not enough evidence to state that the groups were not homogeneous. This is in line with [47], where it was found that age is not an important factor when analyzing public opinion on UAVs.

The opinion, however, had an impact on the results, as the respondents with a negative opinion about UAVs gave higher visual pollution scores to the images than the other respondents.

Regarding the respondents' knowledge of UAVs, no clear differences between groups could be seen.

### 3.2.3. Visual Pollution Function

An ANOVA analysis was carried out to understand which factors had the most influence on the results:

- Environment: Surprisingly, this factor—considered the second most important during the expert workshop—had almost no influence on the results. There was a difference in the acceptance ratio, showing that, in general, UAVs would be more tolerable in an urban environment. However, this difference was very small (four percentage units) and therefore, the environment variable was not included in the model.
- Distance: Distance was a factor that clearly influenced the level of visual pollution. In Figure 5, the scores (rank) given for one UAV flying at different distances are shown.
- Number of UAVs: The number of UAVs significantly increases visual pollution according to the results, see Figure 6.
- Purpose: There were no significant differences in the results regarding the purpose of the UAVs. However, the images with EMS UAVs received slightly lower scores than those without EMS UAVs.
- Awareness: In four images, there was extra information about the UAV, which was supposed to decrease visual pollution, familiarizing people with the UAV. However, the result was the opposite. The mean score for the images that had no information was 3.02, and the mean for those with additional information was 4.01. One possible explanation is that the questionnaire design was not optimal, and the respondents did not correctly understand the meaning of the information squares.

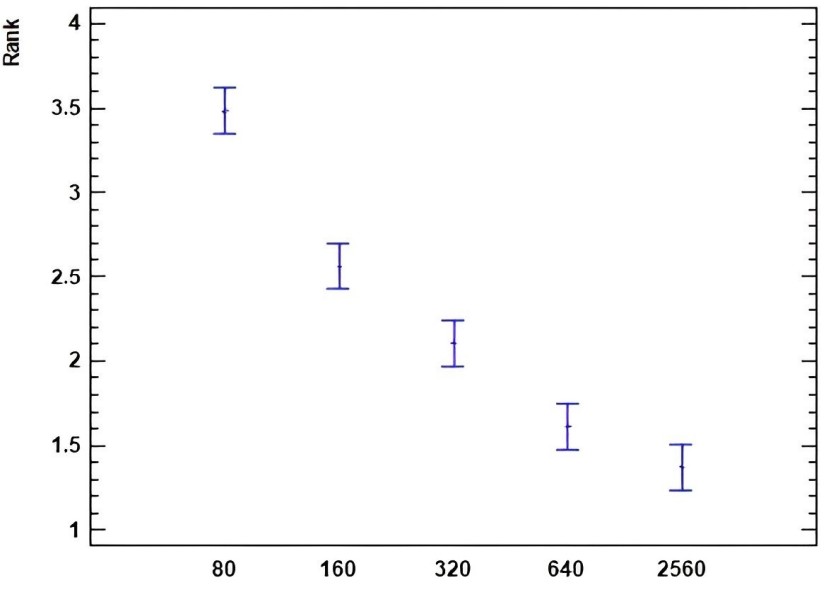

**Figure 5.** Means plot 95% least significant difference intervals that represent the rank of the same UAV placed at different distances.

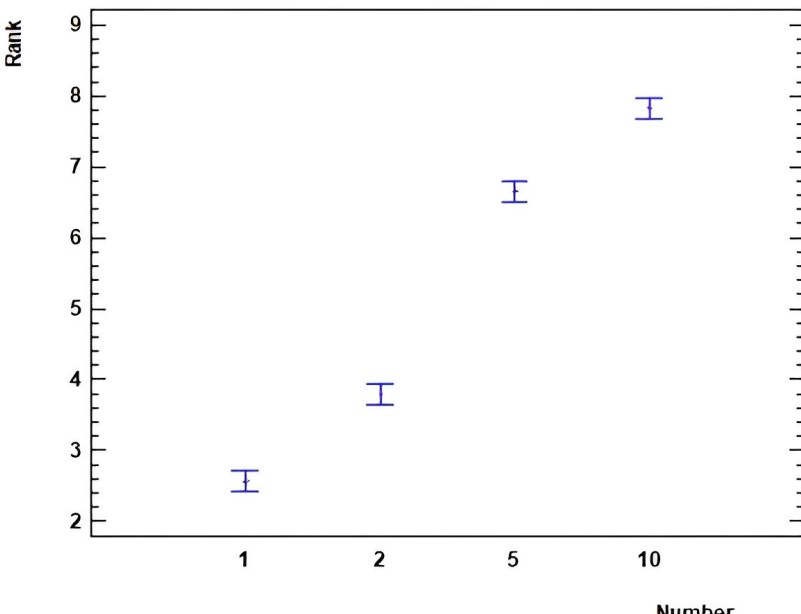

**Figure 6.** Means plot 95% least significant difference intervals that represent the rank of several pictures that contain several UAVs where the closest one is located at 160 m.

After the ANOVA analysis of each factor, several tests were run to decide which variables to include in the model, establishing that all factors should be included except for the environment. This selection was confirmed by the forward stepwise selection and the backward stepwise selection options provided by the Statgraphics software. Thus, the factors accounted for in the model in order of importance are the Number of UAVs, Distance from the closest UAV, Awareness, and Purpose.

The first approach was to carry out a linear regression. The first important output from this regression was to verify that the four parameters included were significant. This was the case, as they all presented $p$-values lower than 0.05. Then, the $R^2$ factor was calculated and resulted in $R^2_{adjusted} = 35.51\%$, which is quite low. This is because the responses are scattered, and participants might not score in the same way, even if the overall trends were similar. However, the tests proved a clear relationship between the scores and the parameters, and therefore it provided relevant information.

Nevertheless, the linear approach did not appear to be the best fit for the data. Therefore, the two main factors (Distance and Number of UAVs) were analyzed separately, and it was found that both had a similar shape to squared-root or logarithm functions. Hence, a non-linear regression was carried out using Excel. Twenty-one different models were run, and the best model was selected based on characteristics such as the complexity of the model, the number of variables, and the interpretability of the function. Finally, the function selected to calculate visual pollution is:

$$VP = \frac{3.83}{\sqrt{Dist}} + 0.97\sqrt{Num} + 20.12\sqrt{\frac{Num}{Dist}} - 0.19Purp + 0.89Info \qquad (1)$$

where *Num* is the number of UAVs that can be seen in the image, *Dist* is the distance from the observant to the closest UAV in the image, *Purp* is the purpose of the UAV (1 if it is an EMS UAV, 0 if it is not) and *Info* reflects if there is extra information regarding the UAV (equal to 1 if there is extra information, else 0).

A simplified function is also proposed. As the EMS factor proved not to have much significance, and the impact that the additional information may have been uncertain, the simplified function only accounts for the distance and the number of UAVs:

$$VP = 47.76\frac{Num^{0.65}}{Dist^{0.67}} + 1.37 \qquad (2)$$

During the rest of the paper, Function (2) will be referred to as the simplified model, and Function (1) as the full model.

The characteristics of the full model are visualized in Figures 7 and 8. When a UAV is further away, the visual pollution decreases, and more UAVs induce more visual pollution. However, the decrease in pollution flattens out when the distance increases, so it does not matter much if the distance increases from 1000 m to 1400 m, while it makes a big difference if it increases from 50 m to 450 m. The same applies to the number of UAVs where the difference between 1 and 2 UAVs is greater than between 8 and 9.

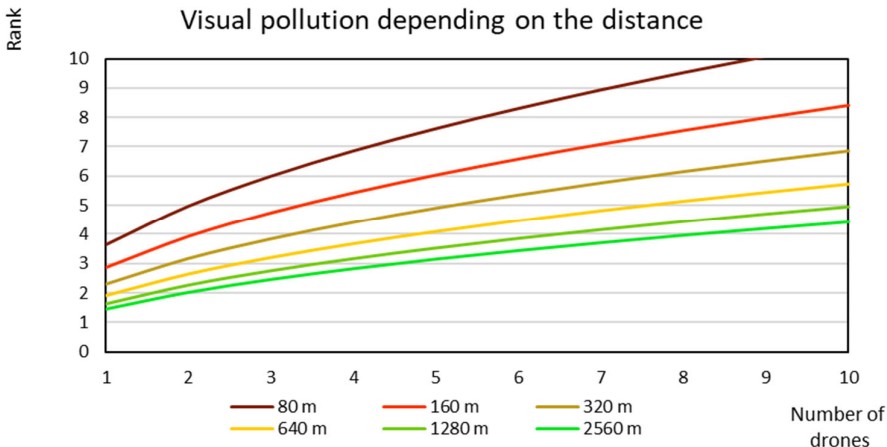

**Figure 7.** Plot of the full visual pollution model depending on the distance.

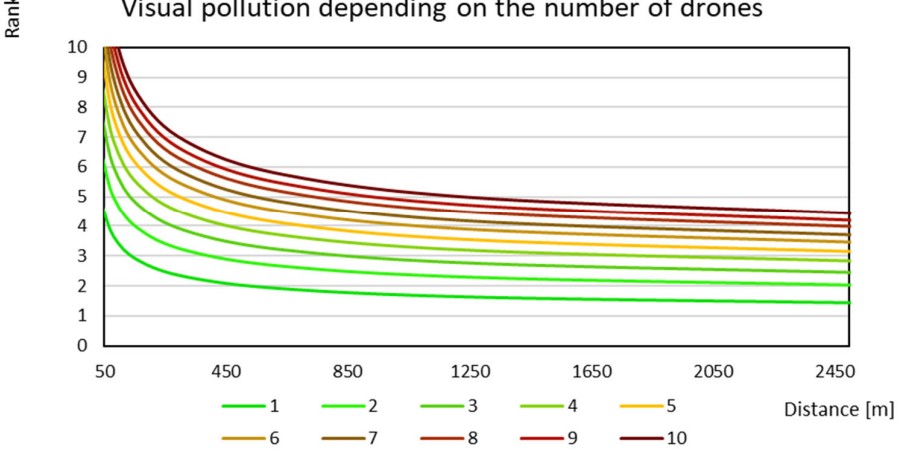

**Figure 8.** Plot of the full visual pollution model depending on the number of UAVs.

The equivalent of Figure 8, but for the simplified model, is shown in Figure 9. The results are very similar, and the simplified model also gives a more logical output when the UAVs are far away, as the visual impact then goes toward the minimum level.

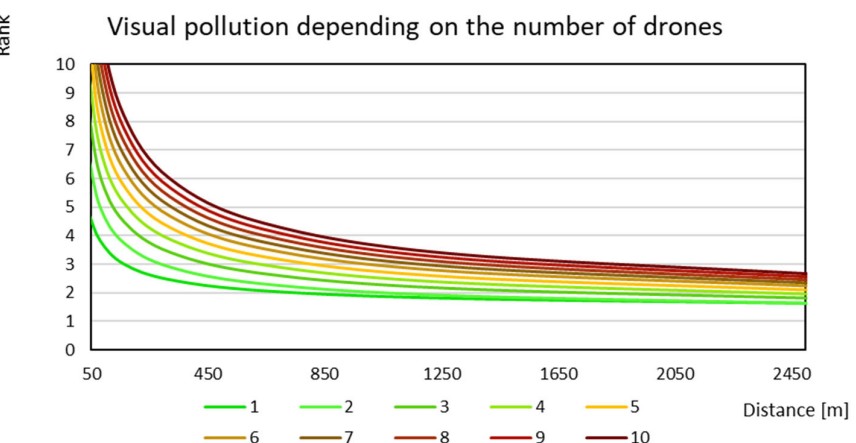

**Figure 9.** Plot of the simplified visual pollution model depending on the distance.

### 3.2.4. Validation of the Results

Images 7, 11, 13, and 14 in the questionnaire (Supplementary File S1 in the Supplementary Material) were not considered when training the model but were meant to be used as validator items. However, 7 and 11 are identical, except for the environment (which is not included in any of the models), and therefore, the models output the same rank for both.

The validation results are shown in Table 3, indicating that the models manage fairly well to predict visual pollution as well as for images not used to train them. The errors for the full model are naturally smaller than for the simplified model. However, since the number of UAVs and the distance to the UAVs are the two dominating factors, the differences are not great. Further analyses were made to understand the accuracy of the models by representing all the values predicted with the models against the average rank of all questions. When calculating the $R^2$ value for the average of all the questions, the result was 96.1% for the complete model and 91.1% for the simplified model. To get a better understanding of this estimation, the images were sorted from the ones that had a lower score (rank) to those that had a higher score, and the result can be seen in Figure 10. It is clear that the full model offers a better fit, but the simplified model is almost as good and has fewer variables, which makes it easier to use.

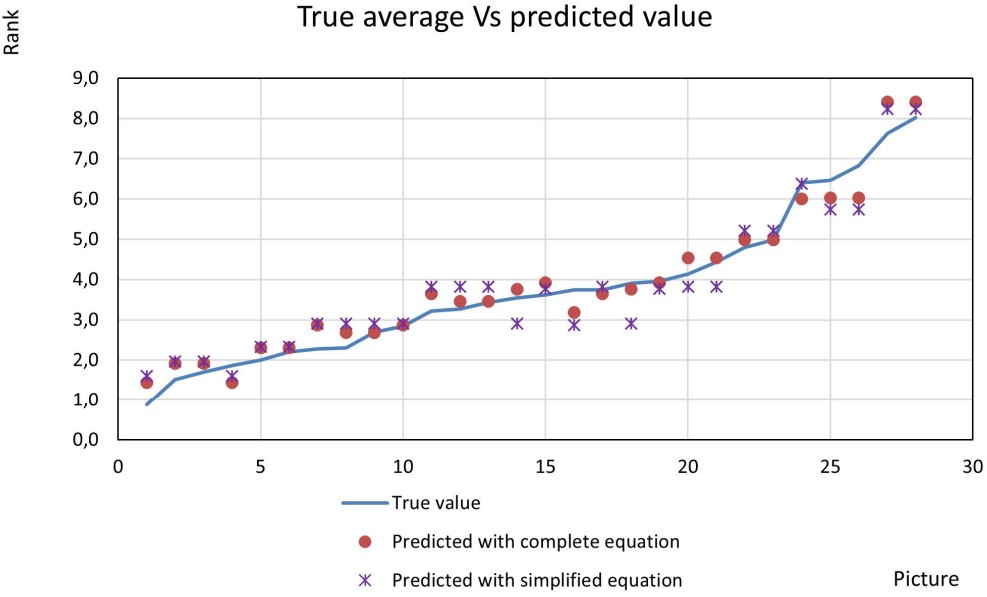

**Figure 10.** The true average compared to the value predicted with the full model and the simplified model.

**Table 3.** Comparative between the real average score provided by the respondents and the score output from both models.

| Image | Distance [m] | Number of UAVs | True Average | Predicted Complete Model | Error [%] | Predicted Simplified Model | Error [%] |
|---|---|---|---|---|---|---|---|
| 7 | 80 | 2 | 4.8 | 5.0 | 3.9 | 5.2 | 8.8 |
| 11 | 80 | 2 | 5.0 | 5.0 | 0.1 | 5.2 | 4.6 |
| 13 | 320 | 2 | 3.7 | 3.2 | 15.0 | 2.9 | 23.1 |
| 14 | 80 | 3 | 6.4 | 6.0 | 6.1 | 6.4 | 0.2 |

## 4. Discussion

In this paper, we present a first attempt to calculate visual pollution from UAVs using mathematical modeling. The functions produced, (1) or (2), can primarily be used to compare different scenarios with each other. For instance, it is possible to calculate the visual pollution induced by a set of UAVs flying specific paths over an area. This can be compared to a scenario when the same UAVs fly different paths, giving less or more visual pollution. To make these calculations possible, some assumptions must be made regarding how to do spatial and temporal discretization and how to summarize the pollution values over the routes considering the estimated affected population. These are aspects that were identified in the expert workshop as important for visual pollution, but they could not be included in the image-based questionnaire. Thus, it would be valuable to complement this work with additional studies focusing on the missing aspects such as movement, noise, and temporal factors.

Furthermore, path planning just taking into account visual pollution would not be efficient. There are other aspects that should be considered; for instance noise pollution, ground and air risk, travel time, and range limitations (see e.g., [48–50]. However, while these other aspects have been studied previously, and models for calculating them exist, this has not been the case for visual pollution. However, it is still an open question on how to include all aspects in a multi-objective path planning model.

The results from the questionnaire were mixed between expected and unexpected. That visual pollution increases with the number of UAVs and when the distance to the UAV decreases can be regarded as trivial. However, the questionnaire data gave the possibility to fit a mathematical function to these factors, making it possible to put a number on the visual pollution. Also, the non-linear relationship between these factors and visual pollution is useful from a planning perspective. For instance, it implies that flight corridors for UAVs would be less visually polluting as this would cluster the UAVs closer together, compared to allowing free path planning.

While knowing that it is an EMS UAV does not seem to lessen the perceived visual pollution, the results indicate that the respondents are more likely to accept a higher level of visual pollution from UAVs used in emergency situations. This result is in line with previous research, e.g., [51], which shows that people have a higher level of acceptability towards medical use cases for UAVs than for other purposes.

The results concerning the environment were surprising, as we expected the respondents to find UAVs in a rural setting more polluting than UAVs in an urban setting. However, similar to the results for EMS UAVs, people seem more accepting of visual pollution in an urban environment. If using the developed visual pollution functions in multi-objective path planning, it may thus be necessary to include some components for the environment. Otherwise, an algorithm would always strive to route a UAV away from highly populated, urban areas, as both visual and noise pollution as well as ground risk increase with the number of affected people on the ground.

An expected result was that respondents with a negative opinion about UAVs gave higher visual pollution scores. This indicates that when working toward a future large-scale implementation of urban air mobility, it is important to take public opinion into account.

While visual pollution is not one of the top concerns for the general public, it is likely that all negative aspects that arise from an increased UAV presence are correlated. Thus, anyone worried about integrity will also perceive UAVs as visually polluting. Therefore, it may be important to improve the general opinion about UAVs, e.g., by early implementation of use cases with high public acceptance, such as emergency medical services.

## 5. Conclusions

In this work, we identified and ranked factors that contribute to the visual pollution produced by UAVs. Using an image-based questionnaire, we also tested the significance and established the relationship between a selection of these factors. While some results were intuitive, e.g., that visual pollution increases with the number of UAVs, other results were not. For instance, we could not see that the environment matters much, indicating that a UAV is equally visually polluting in the city center as in a rural area.

Using the visual pollution scores obtained from the questionnaire, we developed visual pollution functions that can be used to calculate the visual pollution produced by one or several UAVs. These can be used as input in strategic infrastructure planning, design of U-space services, or when doing tactical or operational path planning for UAVs.

Possible future research directions include studying how temporal and dynamic components (e.g., how the UAV moves through the air) affect visual pollution, or how noise and visual pollution interact; something that could not be achieved using an image-based questionnaire. Another interesting area is developing path planning algorithms capable of doing trade-offs between visual pollution, noise pollution, risk, efficiency, and other relevant factors.

**Supplementary Materials:** The following supporting information can be downloaded at: https://www.mdpi.com/article/10.3390/drones7060396/s1, File S1: Questionnaire. File S2: Questionnaire responses.

**Author Contributions:** Conceptualization, K.T. and T.A.G.; methodology, K.T. and T.A.G.; software, K.T.; validation, K.T. and T.A.G.; formal analysis, K.T. and T.A.G.; resources, T.A.G.; data curation, K.T.; writing—original draft preparation, K.T. and T.A.G.; writing—review and editing, K.T. and T.A.G.; visualization, K.T.; supervision, T.A.G.; project administration, T.A.G. All authors have read and agreed to the published version of the manuscript.

**Funding:** This project has received funding from the European Union's Horizon 2020 research and innovation program under grant agreement No. 101006601.

**Data Availability Statement:** The data that was collected using the questionnaire can be downloaded as Supplementary Material.

**Acknowledgments:** We would like to thank the participants in the expert workshop as well as everybody else in the AiRMOUR project who has contributed in some way.

**Conflicts of Interest:** The authors declare no conflict of interest.

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
