# Peer review of "Quantifying Visual Pollution from Urban Air Mobility"

_drones, doi:10.3390/drones7060396_

Round 1

Reviewer 1 Report

TITLE:

Quantifying visual pollution from urban air mobility

SYNOPSIS:

This manuscript considers the problem of quantifying visual pollution caused by UAVs and its impact on emergency response and disaster management. The authors develop a quantitative measure to calculate visual pollution through expert workshops and questionnaires. The results reveal that the number of UAVs and their distances to observers are the main factors contributing to visual pollution. Additionally, the study suggests two visual pollution functions that can aid in path planning for UAVs.

I have read this paper with interest. The topic of quantifying visual pollution is interesting. I have a few recommendations to the authors to further improve the quality of the manuscript:

 - In the introduction, I’d like to see more specific and concrete real-world examples in which noise and/or visual pollution caused by drones has a significant impact on the daily lives of people.

- In the materials and methods section you should refer to Saaty’s early work when mentioning you implemented AHP.

- The quality of figures can be improved since it is hard to distinguish line colors that slight differ in the same graph.

- I recommend providing a more detailed explanation of the questionnaire's design.

- You end the paper with a discussion. I suggest adding a conclusion section which summarizes your work and lists the main takeaways. Also, you need to discuss potential future studies related to your findings.

Author Response

I have read this paper with interest. The topic of quantifying visual pollution is interesting. I have a few recommendations to the authors to further improve the quality of the manuscript:

Thank you so much for reading and commenting. We believe that your comments have helped improve the paper significantly!

 - In the introduction, I’d like to see more specific and concrete real-world examples in which noise and/or visual pollution caused by drones has a significant impact on the daily lives of people

While real world examples are difficult to find, since UAM still is not implemented to a large extent, we added some sentences (and references) regarding health issues related to noise and visual pollution.

- In the materials and methods section you should refer to Saaty’s early work when mentioning you implemented AHP.

Of course, that is appropriate. The first reference after introducing AHP is now to Saaty.

- The quality of figures can be improved since it is hard to distinguish line colors that slight differ in the same graph.

We assume that this relates to Figure 6, 7 and 8 (7, 8, 9 in the revised version). We have made new images with better color contrasts.

- I recommend providing a more detailed explanation of the questionnaire's design.

We are sorry, but we are not sure if the reviewer means explanation of the final design, or explanation of the design choices. If the former, we feel that including the full questionnaire in the supplementary material should be enough (we are not sure however if the supplementary material was sent to the reviews). If the latter, we would need some more details about which parts of the design choices that are unclear, as we believe that we already spent quite a lot of space on this in the manuscript.

- You end the paper with a discussion. I suggest adding a conclusion section which summarizes your work and lists the main takeaways. Also, you need to discuss potential future studies related to your findings.

Added!

Reviewer 2 Report

Dear Authors

Initially, I congratulate you on the research proposal presented.

This paper is about research that aims to understand the visual pollution caused as a negative effect in the use of unmanned vehicles. The authors developed a quantitative measure that can be used to calculate the visual pollution of one or several UAVs. The article is very interesting and pertinent and makes a significant contribution to the understanding of the topic, since it faces the negative issues generated by the use of UAVs. The authors indicate that they used the AHP method as part of the methodology. The work is very well elaborated, however I suggest that some improvements should be made, which I will suggest as follows:

1) The introduction is consistent. The authors made the literature review in the introduction, what I think is very perfect. However, I suggest that they highlight in the introduction the problematic issue of the article and its objectives. Do not subdivide the introduction in sections 1.1 and 1.2. The text needs to be fluid for the reader. 

2) I suggest inserting at the end of the introduction a short paragraph summarizing the other sections of the paper.

3) In relation to the methodology, I suggest illustrating it with a scheme, so that the understanding is visual for the reader. Then, describe it in such a way that other researchers can reproduce the research.

4) The authors talk superficially about the AHP method, I suggest they insert a brief section where they describe it to the reader and its main applications. I also suggest that they insert a justification for choosing the AHP method in detriment of other multicriteria methods, in this sense I suggest reading the text: "Are MCDA Methods Benchmarkable? A Comparative Study of TOPSIS, VIKOR, COPRAS, and PROMETHEE II Methods"; "When is a Decision-Making Method Trustworthy? Criteria for Evaluating Multi-Criteria Decision-Making Methods"; "A Comprehensive Review of the Novel Weighting Methods for Multi-Criteria Decision-Making" ,and "Applying Cocoso, Mabac, Mairca, Eamr, Topsis and Weight Determination Methods for Multi-Criteria Decision Making in Hole Turning Process".

5) In the figures, insert the sources from which they were obtained.

6) In the Links inserted along the text, I suggest inserting the access date.

7) I suggest inserting in the text how was the selection of the people who answered the questionnaire.

I wish you a good review.

Reviewer

Author Response

This paper is about research that aims to understand the visual pollution caused as a negative effect in the use of unmanned vehicles. The authors developed a quantitative measure that can be used to calculate the visual pollution of one or several UAVs. The article is very interesting and pertinent and makes a significant contribution to the understanding of the topic, since it faces the negative issues generated by the use of UAVs. The authors indicate that they used the AHP method as part of the methodology. The work is very well elaborated, however I suggest that some improvements should be made, which I will suggest as follows:

Thank you so much for reading and commenting. We believe that your comments have helped improve the paper significantly!

1) The introduction is consistent. The authors made the literature review in the introduction, what I think is very perfect. However, I suggest that they highlight in the introduction the problematic issue of the article and its objectives. Do not subdivide the introduction in sections 1.1 and 1.2. The text needs to be fluid for the reader.

We removed the subsections and moved the aim and contributions to the end of the section. We also made the aim in italic, to highlight it and make it easier to find.  

2) I suggest inserting at the end of the introduction a short paragraph summarizing the other sections of the paper.

We added such a short paragraph.

3) In relation to the methodology, I suggest illustrating it with a scheme, so that the understanding is visual for the reader. Then, describe it in such a way that other researchers can reproduce the research.

We added a figure illustrating the methodology, and added some information in the bullet list after the figure that should make it easier to follow the steps.

4) The authors talk superficially about the AHP method, I suggest they insert a brief section where they describe it to the reader and its main applications. I also suggest that they insert a justification for choosing the AHP method in detriment of other multicriteria methods, in this sense I suggest reading the text: "Are MCDA Methods Benchmarkable? A Comparative Study of TOPSIS, VIKOR, COPRAS, and PROMETHEE II Methods"; "When is a Decision-Making Method Trustworthy? Criteria for Evaluating Multi-Criteria Decision-Making Methods"; "A Comprehensive Review of the Novel Weighting Methods for Multi-Criteria Decision-Making" ,and "Applying Cocoso, Mabac, Mairca, Eamr, Topsis and Weight Determination Methods for Multi-Criteria Decision Making in Hole Turning Process".

Thank you for all the literature tips! We have now added a longer description of AHP and how we use it, together with a justification for choosing it.

5) In the figures, insert the sources from which they were obtained.

Done!

6) In the Links inserted along the text, I suggest inserting the access date.

Done!

7) I suggest inserting in the text how was the selection of the people who answered the questionnaire.

The following description was already in the manuscript: “No specific selection of the respondents was made. We wanted a general citizen participation so the foremost approach for distributing the survey was snowball sampling. Colleagues, friends and relatives were asked to fill out and spread the survey. This was complemented by website postings (e.g. on http://airmour.eu), email-lists and social media postings on e.g. LinkedIn, Instagram, Reddit and Facebook. The survey was available in English, Spanish, French, Swedish, German, Dutch, Norwegian and Finnish, and was open for 20 days.” We added an additional short note on snowball sampling close to the method figure.

Round 2

Reviewer 2 Report

Dear Authors

I congratulate you for the extensive review work done on the manuscript drones-2406897. I see that you have implemented the suggestions made by the reviewers. I believe that the manuscript is now ready to be considered for publication. I also observed that the insertion of the references is not in the MDPI standard. I believe that the editorial team will determine the proofreading. It is the orientation to follow the MDPI author's guide.

Good luck,

Reviewer

Author Response

Fixed references.